# How Automated Techniques Ease Functional Assessment of the Fetal Heart: Applicability of MPI+™ for Direct Quantification of the Modified Myocardial Performance Index

**DOI:** 10.3390/diagnostics13101705

**Published:** 2023-05-11

**Authors:** Jann Lennard Scharf, Christoph Dracopoulos, Michael Gembicki, Amrei Welp, Jan Weichert

**Affiliations:** Department of Gynecology and Obstetrics, Division of Prenatal Medicine, University Hospital of Schleswig-Holstein, Campus Lübeck, Ratzeburger Allee 160, 23538 Lübeck, Germany; christoph.dracopoulos@uksh.de (C.D.); michael.gembicki@uksh.de (M.G.); amrei.welp@uksh.de (A.W.); jan.weichert@uksh.de (J.W.)

**Keywords:** cardiac function, myocardial performance index, MPI, Tei index, automation, artificial intelligence

## Abstract

(1) Objectives: In utero functional cardiac assessments using echocardiography have become increasingly important. The myocardial performance index (MPI, Tei index) is currently used to evaluate fetal cardiac anatomy, hemodynamics and function. An ultrasound examination is highly examiner-dependent, and training is of enormous significance in terms of proper application and subsequent interpretation. Future experts will progressively be guided by applications of artificial intelligence, on whose algorithms prenatal diagnostics will rely on increasingly. The objective of this study was to demonstrate the feasibility of whether less experienced operators might benefit from an automated tool of MPI quantification in the clinical routine. (2) Methods: In this study, a total of 85 unselected, normal, singleton, second- and third-trimester fetuses with normofrequent heart rates were examined by a targeted ultrasound. The modified right ventricular MPI (RV-Mod-MPI) was measured, both by a beginner and an expert. A calculation was performed semiautomatically using a Samsung Hera W10 ultrasound system (MPI+™, Samsung Healthcare, Gangwon-do, South Korea) by taking separate recordings of the right ventricle’s in- and outflow using a conventional pulsed-wave Doppler. The measured RV-Mod-MPI values were assigned to gestational age. The data were compared between the beginner and the expert using a Bland-Altman plot to test the agreement between both operators, and the intraclass correlation was calculated. (3) Results: The mean maternal age was 32 years (19 to 42 years), and the mean maternal pre-pregnancy body mass index was 24.85 kg/m^2^ (ranging from 17.11 to 44.08 kg/m^2^). The mean gestational age was 24.44 weeks (ranging from 19.29 to 36.43 weeks). The averaged RV-Mod-MPI value of the beginner was 0.513 ± 0.09, and that of the expert was 0.501 ± 0.08. Between the beginner and the expert, the measured RV-Mod-MPI values indicated a similar distribution. The statistical analysis showed a Bland-Altman bias of 0.01136 (95% limits of agreement from −0.1674 to 0.1902). The intraclass correlation coefficient was 0.624 (95% confidence interval from 0.423 to 0.755). (4) Conclusions: For experts as well as for beginners, the RV-Mod-MPI is an excellent diagnostic tool for the assessment of fetal cardiac function. It is a time-saving procedure, offers an intuitive user interface and is easy to learn. There is no additional effort required to measure the RV-Mod-MPI. In times of reduced resources, such assisted systems of fast value acquisition represent clear added value. The establishment of the automated measurement of the RV-Mod-MPI in clinical routine should be the next level in cardiac function assessment.

## 1. Introduction

A functional cardiac assessment of a fetus using echocardiography has become increasingly important. It allows for the early detection of subclinical cardiac dysfunction and reveals intrauterine functional cardiac remodeling, and can therefore contribute to improving and predicting perinatal outcomes [1,2,3]. The myocardial performance index (MPI, synonym: Tei index) was initially proposed for adult cardiology and can be used to assess the left as well as the right ventricular cardiac function [4]. More recently, this Doppler index has also been increasingly applied to fetal hearts and has shown promise in the assessment of right ventricular (RV) fetal cardiac function [5,6,7,8,9,10]. The MPI has been developed continuously since (Mod-MPI) by using valve clicks as landmarks to improve inter- and intra-observer agreements and, thus, make the measurement more reliable and reproducible [11,12,13,14,15]. The MPI is the ratio of the isovolumetric to ejection time of cardiac intervals [4,16]. As a non-invasive, pulsed-wave (PW) Doppler-derived measure of global myocardial function, it is currently used to evaluate fetal cardiac anatomy, hemodynamics and function [1]. By now, the Mod-MPI has been widely used to assess the function of the heart in a wide variety of pathological intrauterine conditions by estimating early fetal cardiac adaptive changes in complicated pregnancies, including fetal growth restriction (FGR), gestational diabetes, twin-to-twin transfusion syndrome (TTTS), congenital heart defects (CHD), pre-eclampsia, intrahepatic cholestasis of pregnancy (ICP) and other fetal intra- and extracardiac conditions [1,3,17,18,19,20,21,22]. Ventricular dysfunction is associated with higher MPI values. The isovolumetric relaxation time (IRT), being the main MPI parameter, is often prolonged even in the very early stages of cardiac dysfunction, as the fetal myocardium requires more time for its relaxation during diastole. Accordingly, an increased IRT is accompanied by a reduced ejection time (ET) and, therefore, an abnormal increase in MPI. This indicates cardiac dysfunction, whereas in general, different components may be affected [20]. Fetal circulation is right-heart-dominant and alterations in cardiac function can demask earlier on this side. Therefore, the RV-Mod-MPI is considered an important parameter in particular, even though the published literature has focused mainly on the investigation of the LV-Mod-MPI so far. The calculation of the RV-Mod-MPI in advanced gestational ages (GAs) requires the acquisition of two different anatomical planes in two different cardiac cycles with different fetal heart rates, because the tricuspid and pulmonary valves diverge at distinct anatomical levels. Due to this anterior displacement of the pulmonary valve (synonym: pulmonary–tricuspid discontinuity), which develops between the 20th and 26th weeks of gestation, the isovolumetric times (isovolumetric contraction time (ICT) and IRT) on the right side of the heart cannot be obtained from the same cardiac cycle individually [1,11,14,17,20,23,24,25]. Consequently, it is suggested that the accuracy of the RV-Mod-MPI could be more influenced and possibly compromised by variations in the fetal heart rate. There is a wide variation in the quoted reference values of the MPI to date—which are still inconsistent—ranging from 0.35 to 0.60 as the mean values. Therefore, a significant correlation between the MPI and the gestational age (GA) is controversially debated in the current literature [1,8,15,18,20,23,26,27,28,29,30,31,32,33]. The Mod-MPI is less dependent on factors that impair image acquisition and quality than the other methods used for assessing fetal cardiac function (e.g., maternal BMI, anterior placenta, oligohydramnios), is less dependent on fetal anatomy and position or precise imaging and, by incorporating only time intervals, is less prone to artifacts such as fetal movements.

An ultrasound examination is highly examiner-dependent—especially in filigreed, moving and changing fetal structures—and training is of enormous significance, even in the acquisition of the MPI [34,35]. Future experts will progressively be guided by applications of artificial intelligence (AI), on whose algorithms prenatal diagnostics will rely on increasingly [36,37,38,39,40,41]. By now, the acquisition of the MPI can also be achieved automatically using AI [23,30,42,43,44]. The aim of this study was to demonstrate the feasibility of whether less experienced operators might benefit from an automated tool of MPI quantification in the clinical routine.

## 2. Materials and Methods

### 2.1. Subjects

In this prospective study, a total of 85 unselected, normal, singleton, second- and third-trimester fetuses with normofrequent heart rates were examined during a targeted ultrasound survey. All women were routinely investigated by the application of MPI+™ between August 2021 and February 2023. The used RV-Mod-MPI values were acquired separately by a beginner investigator (J.L.S.) and an expert investigator (J.W.) based on consecutively, but independently, recorded ultrasound images, and the values had to fulfill predefined in- and exclusion criteria. Fetuses with structurally abnormal hearts were excluded, as were hearts with tachycardia or fetuses with target structures that could not be clearly identified and accessed or fetuses with poor Doppler image acquisition. Informed consent was obtained from all participants. The measured RV-Mod-MPI values were matched with those reported in the literature.

### 2.2. Acquisition of PW Doppler Waveform Images

As a pre-requisite for calculating the RV-Mod-MPI, separate recordings of the right ventricular in- and outflow blood velocity waveforms using a conventional PW Doppler were recorded using a Samsung Hera W10 ultrasound device equipped with transabdominal probes 3–10 and 1–8 MHz (S-Vue™-Transducer CA3-10A and CV1-8A) (MPI+™, Samsung Healthcare, Gangwon-do, Republic of Korea). The measurement was performed as described below with the following predefined ultrasound settings: The Doppler sweep velocity was kept at 5 to 10 cm/s, the gain was optimized to visualize the valve clicks clearly and the wall motion filter was set at 200 Hz. The fetal heart occupied 75% of the screen, corresponding to an appropriate image magnification. The sample volume was placed across the tricuspid valve (approx. 6 mm) using an apical four-chamber view, and subsequently across the pulmonary valve (approx. 2 mm) in either the short axis view or sagittal plane. The angle of insonation was less than 15–20°. The measurements of both the in- and outflow took place within a few seconds consecutively without a significant time difference in the acquisition of the relevant target structures and without a significant difference in fetal heart rate. A difference of ≤5 beats/min between the RV in- and outflow was considered appropriate and included in the study (Figure 1, Appendix A).

### 2.3. Application of MPI+™

For the calculation of the RV-Mod-MPI and its components (ICT, IRT, and ET), an analysis of the measured PW Doppler waveform signals of the in- and outflow of the right ventricle was acquired semiautomatically by alignment and synchronization based on pulmonary valve closure clicks in both separate images using the built-in software MPI+™, installed on a high-resolution ultrasound machine, by the same investigators mentioned above. The individual stepwise instructions of the program were followed (Figure 1, Appendix A). A priori, despite the well-known wide variation in MPI reference values, we define narrow limits for clinical practice with a difference of <0.05 as excellent, <0.1 as good, and <0.2 as satisfying.

### 2.4. Statistics

All the semiautomatically derived RV-Mod-MPI values were assigned to GA descriptively. The data were compared between the beginner and expert using a Bland-Altman plot (average between the RV-Mod-MPI of the beginner and expert against the difference between these two) to test the agreement between both operators. The intraclass correlation coefficient (ICC) was calculated for inter-rater reproducibility. ICC values were calculated using a two-way mixed effects model for absolute agreement. GraphPad Prism 9 for Mac (version 9.4.1, GraphPad Software Inc., La Jolla, CA, USA), SPSS Statistics (version 28.0.1.0, IBM Corporation, Armonk, NY, USA) and Microsoft Excel 2016 for Mac (Version 16.66.1, Microsoft Corp., Redmond, WA, USA) were used.

## 3. Results

The mean maternal age was 32 years (range of 19 to 42 years), and the mean pre-pregnancy maternal body mass index (BMI) was 24.85 kg/m^2^ (ranging from 17.11 to 44.08 kg/m^2^). The mean gestational age (GA) was 24.44 weeks (ranging from 19.29 to 36.43 weeks) (Table 1). The RV-Mod-MPI could be assessed for all the included women for both operators. The average RV-Mod-MPI value of the beginner was 0.513 ± 0.09, and that of the expert was 0.501 ± 0.08 (Appendix A). The RV-Mod-MPI values corresponded with the expected distribution pattern and increased with GA, as shown by the linear regression in Figure 2. Between the beginner and expert, the measured RV-Mod-MPI values indicated a similar distribution (Figure 3). The statistical analysis showed a Bland-Altman bias of 0.01136 (standard deviation (SD) of bias: 0.09122; 95% confidence interval (CI) from −0.00823 to 0.03095) and 95% limits of agreement from −0.1674 (LoA−, 95% CI from −0.2014 to −0.1335) to 0.1902 (LoA+, 95% CI from 0.1562 to 0.2241) (Figure 4). A total of 75.30% showed good agreement (difference of <0.1), with 45.88% of measurements even showing excellent agreement (difference of <0.05). The intraclass correlation coefficient (ICC) was 0.624 (95% CI from 0.423 to 0.755).

## 4. Discussion

Our study clearly demonstrates that less experienced operators could benefit from an automated tool for MPI quantification in routine clinical practice. For experts as well as for beginners, the RV-Mod-MPI is an excellent diagnostic tool for the assessment of fetal cardiac function. In total, there was no significant difference between the semiautomated measurements of the beginner and expert.

Contrary to the statement by Mahajan et al. in 2014, our findings suggest that the assessment of the MPI does not require specific expertise and can just as easily be measured by a beginner with only minimal theoretical knowledge [20]. The measurement appears to be no longer technically challenging, even though its simple application was already propagated 20 years ago [8,12]. Through the standardization of the application process using AI (software tool MPI+), its measurement has been simplified and is reproducible. Based on our data, there is no limitation to its transition into clinical practice anymore.

In 2011, it could already be shown that the manual measurement of the fetal MPI seemed to be no more challenging than the measurement of other fetal ultrasound parameters. The study by Cruz-Martínez et al. is one of the few to evaluate the learning curve required for an inexperienced operator to yield reproducible MPI measurements manually, not AI-based. They reported that, on average, adequate practical skills in the acquisition of fetal MPI measurements by an inexperienced operator was achieved after 65 ultrasound examinations [34]. The findings of this present study contradict prior research. From the beginning, the RV-Mod-MPI values acquired by the two operators were already very similar, as shown in Appendix A with the values listed chronologically. The acquisition time was usually well below the generally assumed average of 2.5 min per measurement, so it is a time-saving procedure and it offers an intuitive user interface as well as being easy to learn. There is no additional effort required to measure the RV-Mod-MPI. If anything, the learning curve here turns out to be extremely steep. Of course, a basic theoretical understanding of fetal anatomy, its technical application and its sonomorphological representation is extremely helpful for measuring the RV-Mod-MPI, but is no longer a mandatory requirement to assess the fetal RV-Mod-MPI due to step-by-step instructions provided by the software. Nevertheless, this understanding, as well as knowledge of the white paper, can reduce common pitfalls and optimize value acquisition [16]. However, extensive medical expertise is required for the evaluation, classification and interpretation of its values in a clinical context. Because of the fact that the software works reliably, a loss of competence is not expected in the future, in which experts will progressively be guided by applications of AI.

As a result of the highly heterogeneous measurement techniques used so far, with priority given to the lack of standard criteria regarding caliper placement, clinicians are currently confronted with a wide variation in the quoted reference values for the MPI to date—which still lack consistency—ranging from 0.35 to 0.60 as the mean values. The values of the RV-Mod-MPI for both operators corresponded with this expected distribution pattern and increased with GA (Figure 3), even though a significant correlation between the MPI and GA has been controversially discussed in the current literature [1,8,15,18,20,23,26,27,28,29,30,31,32,33]. The present values of the RV-Mod-MPI were very similar to those of Kang et al., who investigated the clinical value of the MPI+™ tool for the assessment of cardiac function in TTTS. These averaged 0.500 ± 0.08 between 20.00 and 23.60 weeks of gestation [23]. When interpreting the RV-Mod-MPI, it is less important to consider its singular measurement than its course. Measurements of the RV-Mod-MPI by MPI+™ can be performed at nearly all GAs [1,18,20,28]. It is assumed that increasing values of the MPI are correlated with increasing dysfunction of the ventricles [1,8,12]. A direct correlation between increased MPI values and adverse perinatal outcomes has been suggested [21]. Currently, there are no absolute cut-offs that require any medical intervention. Using AI algorithms, the RV-Mod-MPI values should become more reproducible in the future and the lack of standard criteria, which dominated for a long time, is now being circumvented. Appropriately implemented percentiles within the AI software can support a clinician in interpreting the results. Standardized gestation-specific reference ranges will pave the way for its use in clinical routine as another component [1,20,31,32].

Nevertheless, some limitations of the present study must be mentioned. Only two operators acquired the RV-Mod-MPI values. However, these two operators (beginner and expert) were selected representatively of their respective setting as holders of DEGUM (German Society for Ultrasound in Medicine) level 1, the minimum standard of a sonographer, and DEGUM level 3 for gynecology and obstetrics, with the former confirming to be familiar with the basic principles of ultrasound diagnostics and the latter attesting to be a proven expert far beyond the basic knowledge. The examined collective was quite limited, with only 85 unselected, normal, singleton, second- and third-trimester fetuses without CHD in healthy women, focusing on the 25th week of gestation on average. In addition, the time required to acquire this small number of fetuses was rather prolonged.

The acquisition of images for the calculation of the RV-Mod-MPI should be ideally performed without fetal physical or respiratory movements. Knowing the white paper and following its aspects are worth emphasizing in particular [16]. The angle of insonation should be as small as possible, ideally <15°. The software adjusts the heart rate between the in- and outflow automatically, if it is within ≤5 beats/min, and synchronizes the images and places the calipers automatically. Therefore, heart rate fluctuations due to anterior displacement of the pulmonary valve were not as challenging as the acquisition of the relevant target structures without a significant time difference, and therefore, the difference in heart rate between the in- and outflow tracts was as low as possible, preferably less than 5 beats/min. There is no common consensus yet regarding which heart rate discrepancy might influence fetal MPI. It is considered that a discrepancy of >10 beats/min could lead to bias [23,33]. To obtain adequate images, it is important not only to apply stringent criteria, but also to ensure proper machine settings. Indeed, the algorithm behind MPI+™ does not control, and therefore does not automate, the ultrasound pre-settings, although they affect the calculation and repeatability of the measurement of MPI values. It has been proven that pre-settings can influence the acquisition of images. Therefore, standardization is mandatory above all in the monitoring of pregnancies [1,20,45].

Inevitably, alternative methods to overcome the limitation of pulmonary–tricuspid discontinuity, such as attempting a dual gate PW Doppler in the short-axis view for simultaneously recording the in- and outflow as well as valve clicks, become obsolete with the application of AI [23,24].

An assessment of the MPI could be implemented in routine diagnostics quite simply—effectively on-the-fly. The ability of obstetric ultrasonography to detect outflow tract anomalies is increased significantly, if the visualization of the ventricular outflow tract complements the four-chamber view that is well-established in routine diagnostics [46]. As a result of this insight, the outflow tract and the four-chamber view were included in routine screening for CHD, which is evidence-based and consistent with the recent guidelines and recommendations from other international expert associations [47,48]. Its application as a screening as well as a follow-up tool for global cardiac dysfunction makes the MPI a valuable predictor of CHD.

## 5. Conclusions

For experts as well as for beginners, the RV-Mod-MPI is an excellent diagnostic tool for the assessment of fetal cardiac function. It is a time-saving and highly reproducible procedure, offers an intuitive user interface and is easy to learn. Thus, there is no additional effort required to measure the RV-Mod-MPI. In times of reduced resources, such assisted systems of fast value acquisition represent clear added value. Further studies will examine the applicability of the RV-Mod-MPI in abnormal pregnancies with pathological fetal intrauterine conditions such as FGR, gestational diabetes, TTTS, CHD, pre-eclampsia, ICP and other fetal intra- and extracardiac conditions, as well as compare its learnability, feasibility and reliability with other methods of prenatal assessments of fetal cardiac function, such as 2D speckle-tracking imaging. The implementation of automated measurements of the RV-Mod-MPI in clinical routine should be the next level in cardiac function assessment.

## Figures and Tables

**Figure 1 diagnostics-13-01705-f001:**
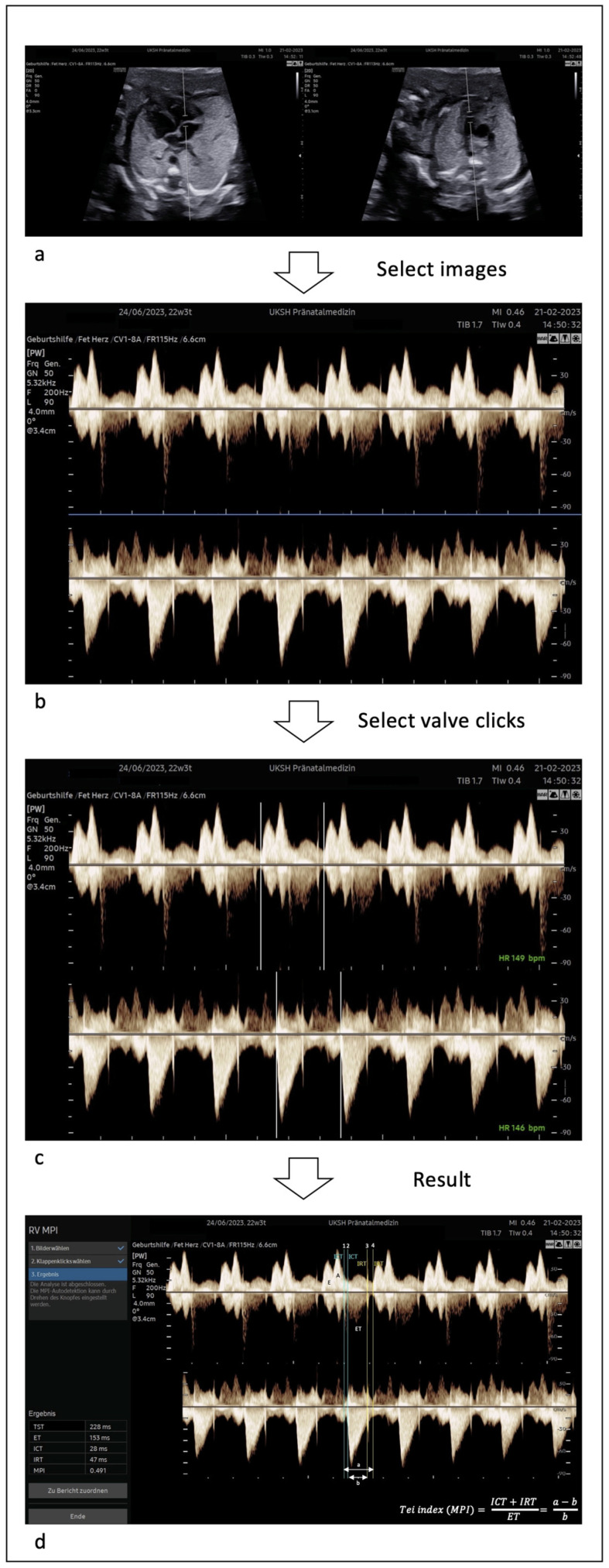
Acquisition of RV-Mod-MPI using MPI+™. (**a**,**b**): Identification and classification of the in- and outflow of the right ventricle (tricuspid and pulmonary valve) by placement of the sample volume by the operator. (**c**): Selection of one cardiac cycle each from the in- and outflow manually, and calculation of the heart rates with synchronization of the in- and outflow images based on the beginning of the pulmonary valve closure clicks by the software automatically. (**d**): Calculation of the RV-Mod-MPI and its components by the software automatically. Upper panel: tricuspid valve, lower panel: pulmonary valve. ICT: isovolumetric contraction time, IRT: isovolumetric relaxation time, ET: ejection time. 1: closure tricuspid, 2: aperture pulmonal, 3: closure pulmonal, 4: aperture tricuspid. Adapted from [16].

**Figure 2 diagnostics-13-01705-f002:**
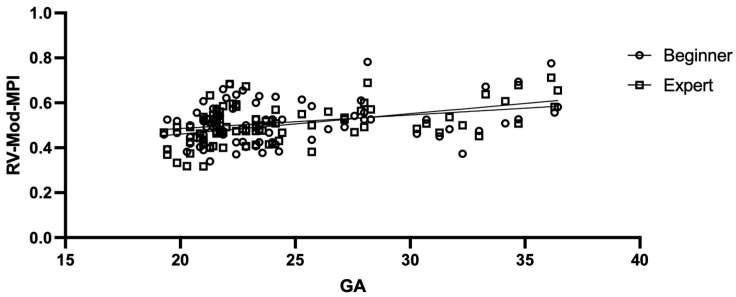
RV-Mod-MPI versus gestational age (GA). *X*-axis: GA in weeks. *Y*-axis: RV-Mod-MPI values of beginner (dot) and expert (square). Continuous line represents linear regression. MPI+™ was performed mainly as part of second-trimester screening, resulting in a clustering of measurement points in this area.

**Figure 3 diagnostics-13-01705-f003:**
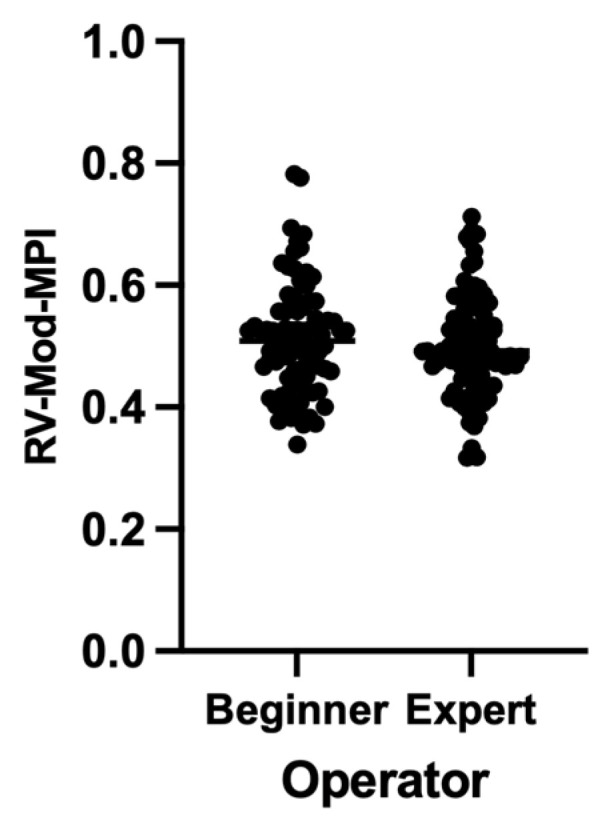
Distribution of the RV-Mod-MPI measured values by beginner and expert.

**Figure 4 diagnostics-13-01705-f004:**
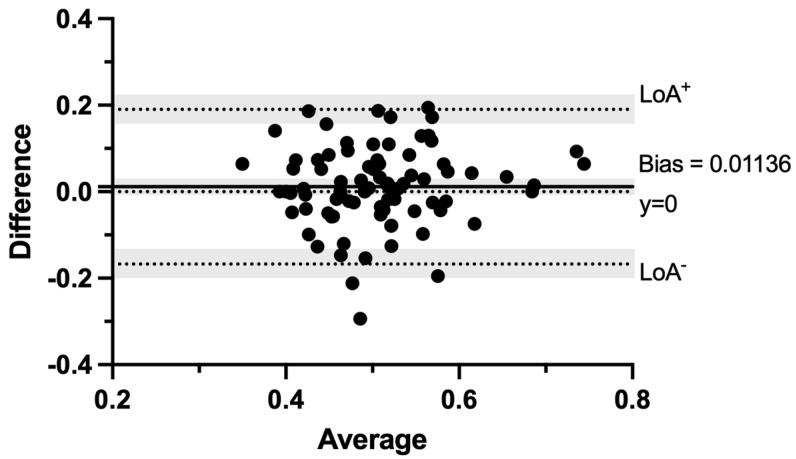
Bland-Altman plot of agreement in RV-Mod-MPI measurements of beginner and expert. *X*-axis: Averages of RV-Mod-MPI values of beginner and expert. *Y*-axis: Differences between the measured values of beginner and expert. Dotted line represents 95% limits of agreement (LoA) and absolute agreement (y = 0) of measurements, and continuous line represents Bland-Altman bias. Shaded areas represent 95% confidence interval (CI) limits for bias and limits of agreement.

**Table 1 diagnostics-13-01705-t001:** Clinical characteristics of the study population (*n* = 85).

Characteristics	Mean (Range)
Maternal age, years	32.86 (19–42)
Nulliparous, %	43.53
Primiparity, %	32.94
BMI prior to pregnancy, kg/m^2^	24.85 (17.11–44.08)
GA at targeted ultrasound, weeks of gestation	24.44 (19+2–36+3)
Fetal cephalic presentation, %	61.18

## Data Availability

The data presented in this study are available from the corresponding author upon request. The data are not publicly available due to data privacy restrictions.

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
