# Peer review of "How Automated Techniques Ease Functional Assessment of the Fetal Heart: Applicability of MPI+™ for Direct Quantification of the Modified Myocardial Performance Index"

_diagnostics, 2023, doi:10.3390/diagnostics13101705_

Round 1

Reviewer 1 Report

This paper aims to evaluate use of MPI+ by different operators in fetal heart analysis.

The paper suffers of some Major Issues should be discussed before considering it for publication in Diagnostics.

MAJOR ISSUES:

1)    It is not clear if both operators acquired two different sets of US scans or they simply measured the same US scan set. Please clarify. If the US images were acquired by a single operator, its level of skillness should be reported. Furthermore, this fact should be reported as a limitation of the study.

2)    Intervals of variation of the bias in Bland-Altman analysis should be reported in order to clearly understand if the bias is statistically different from zero or not.

3)    Authors claim that the less experienced operator could benefit from the use of MPI+. This imply that authors consider the level of agreement between the two operators as good. However, even if the bias is small (not different from zero, likely, see issue #2), the LoA are equal to about 35-40% of the average measurement. Authors should discuss about the maximum value of the LoA that should be considered as acceptable in order to evaluate the agreement as good/satisfying.

4)    Authors validated the agreement in apparently healthy subjects. However, US scans can be noisier in pathological conditions, thus impacting the level of agreement. Accordingly, authors should evaluate the agreement enrolling pathological subjects. Alternatively, they should recognize this point as a limitation of the study.

Author Response

We thank you for your efforts to read our paper and for your valuable comments.

1)
It is not clear if both operators acquired two different sets of US scans or they simply measured the same US scan set. Please clarify. If the US images were acquired by a single operator, its level of skillness should be reported. Furthermore, this fact should be reported as a limitation of the study.

Both operators acquired two different sets of ultrasound images of the same fetal heart.
We have included this aspect in the paragraph method and clarified our wording.

The level of skillness was also mentioned in the discussion (DEGUM I and DEGUM III).

2)
Intervals of variation of the bias in Bland-Altman analysis should be reported in order to clearly understand if the bias is statistically different from zero or not.

The statistical evaluation method suggested by Bland and Altman 1983 addresses issues of estimation rather than significance testing. The question of interpretation of the individual clinical measurement is crucial (Bland, J. Martin, and DouglasG Altman. Statistical methods for assessing agreement between two methods of clinical measurement. The lancet 327.8476 (1986): 307-310.; doi: 10.1002/uog.122).

It is a simple method to assess bias between mean differences and to estimate an agreement interval into which 95% of the differences of the second approach are included compared to the first approach (doi: http://dx.doi.org/10.11613/BM.2015.015).

Quantifying those differences is essential, not testing statistical hypotheses for equality of methods. The Bland-Altman graphical method provides a simple way to graphically represent and quantify the agreement between measurement methods. However, the interpretation of the limits of agreement obtained is a clinical, not a statistical, issue (doi: 10.1055/s-2007-959047).

Assessment of a clinician to evaluate the range between the limits of agreement to conclude that the methods agree sufficiently, is required. It is not a statistical decision (doi: 10.1055/s-2007-959047; doi: 10.1002/uog.122.

It is true that the smaller the sample size, the wider the confidence intervals, which could make the results more uncertain.

So, Bland-Altman plots are generally interpreted informally, without further analyses. We have calculated the standard deviation (SD) of the bias (differences between the two operators) as well as the 95% confidence interval (CI) for the mean (bias) and the limits of agreement (LoAs) and added these values to the paragraph results and to Figure 4, to better understand Bland-Altman (http://dx.doi.org/10.11613/BM.2015.015).

These 95% prediction bands are wider than the 95% limits of agreement and so provide a more accurate prediction of where to expect future differences between the two operators.

3)
Authors claim that the less experienced operator could benefit from the use of MPI+. This imply that authors consider the level of agreement between the two operators as good. However, even if the bias is small (not different from zero, likely, see issue #2), the LoA are equal to about 35-40% of the average measurement. Authors should discuss about the maximum value of the LoA that should be considered as acceptable in order to evaluate the agreement as good/satisfying.

S. comment #2. For better understanding, we added in the paragraph methods a priori values of the RV-Mod-MPI for classification in the clinical context for clinical routine (‘excellent’, ‘good’, ‘satisfying’) and in the paragraph results corresponding percentages.

4)
Authors validated the agreement in apparently healthy subjects. However, US scans can be noisier in pathological conditions, thus impacting the level of agreement. Accordingly, authors should evaluate the agreement enrolling pathological subjects. Alternatively, they should recognize this point as a limitation of the study.

Thank you for your comment. Absolutely correct, the fetuses in our collective were demonstrably healthy, as confirmed by detailed targeted fetal ultrasound. The aim of this study was to examine the feasibility and validity of measuring RV-Mod-MPI using the MPI+ software with intention in a healthy normal collective. Obviously, further studies must follow, specifically on pathological pregnancies such as twin-to-twin transfusion syndrome (TTTS), fetal growth restriction (FGR) or maternal diabetes. Indeed, we measured pathological cases, intentionally, we excluded them from this study. It will be analyzed in further projects.

We added our sentence in line 282: Further studies will examine the applicability of RV-Mod-MPI in abnormal pregnancies with pathological fetal intrauterine conditions such as FGR, gestational diabetes, TTTS, CHD, pre-eclampsia, ICP and other fetal intra- and extracardiac conditions as well as comparing its learnability, feasibility, and reliability with other methods for prenatal assessment of fetal cardiac function, such as 2D speckle-tracking imaging.

We already recognize this point as a limitation in our discussion in line 243: The examined collective was quite limited, with only 85 unselected, normal singleton second and third trimester fetuses without CHD in healthy women, ...

Reviewer 2 Report

This study aimed to determine the viability of using an automated tool for Myocardial performance index (MPI) quantification during fetal cardiac function ultrasound examination. The study included 85 average singleton fetuses with irregular heart rates in their second and third trimesters. Using the MPI+TM tool, a novice, and an expert measured the modified right ventricular MPI (RV-Mod-MPI) on a Samsung Hera W10 ultrasound system using the MPI+TM tool. The study found that the RV-Mod-MPI is an excellent diagnostic tool for fetal cardiac function evaluation with an intuitive user interface that is simple to master. The automated measurement of the RV-Mod-MPI saves time and has significant value, particularly with limited resources. The results demonstrated a high degree of agreement between the measurements made by the novice and the expert, indicating that less experienced operators may benefit from the automated tool of MPI quantification in routine clinical practice. The study suggests incorporating automated RV-Mod-MPI measurement into clinical practice as the next step in embryonic cardiac function evaluation.

Authors are requested to follow up the following comments.

1. Rewrite the abstract in a paragraph format with a precise problem statement, objective, technical method, results, and summarized conclusions.

2. more citations should be included to discuss the impact of noise removal on the index. The following articles may be helpful: https://doi.org/10.1080/21681163.2013.875486 and DOI: 10.1109/TITB.2004.832545

3. A review of other indices may be helpful in the introduction.

4. The title of Figure 1 should be shortened, and the description can be given within the text.

5. A flow diagram for the algorithm would be necessary to explain the method.

6. Results should be tabulated and compared with others from the literature.

7. It is recommended to use non-linear fitting to the data in the other figures and compare its accuracy, precision, and sensitivity with the proposed ones https://doi.org/10.3390/bioengineering10020249

7. The contribution of the work should be highlighted in the discussion and why it is worth publishing.

8. conclusions should be further explored, and a vivid picture of future work is necessary.

Author Response

We thank you for your efforts to read our paper and for your valuable comments.

1. Rewrite the abstract in a paragraph format with a precise problem statement, objective, technical method, results, and summarized conclusions.

Writing the abstract, we followed the guidelines of the journal MDPI diagnostics:
https://www.mdpi.com/journal/diagnostics/instructions - the structure is given: Background, Methods, Results, Conclusion.

2. more citations should be included to discuss the impact of noise removal on the index. The following articles may be helpful: https://doi.org/10.1080/21681163.2013.875486 and DOI: 10.1109/TITB.2004.832545

The predefined echocardiography ultrasound settings, e.g. gain, were optimized to visualize the valve clicks clearly. Their clear visualization is crucial for the calculation of the RV-Mod-MPI and was optimized by the described pre-settings. The noise removal is not the crucial parameter for the application of the algorithm. Therefore, we decided to keep the paragraph 'methods' with its execution of the ultrasound specifications as it is.

3. A review of other indices may be helpful in the introduction.

Yes, you are right, other indices like E/A ratio (diastolic function) or the annular displacement (systolic function) are also known cardiac parameter, but in this paper, we focused on the global function of the fetal heart as a usable, reproducible parameter. In the field of fetal echocardiography, Mod-MPI is an easy key index for the assessment of global cardiac function. Of course, during the measurement of MPI, the biphasic wave of the tricuspid valve is also registered. Alternatively, two-dimensional speckle tracking (2D-STE) with multiple parameters is used, which can also assess segmental cardiac function. Intentionally, we did not include other indices, because they were not in the scope of this paper. But indeed, other parameters can also give separate information about diastolic and systolic function. However, this requires additional settings and calculations and were therefore not mentioned in the introduction. From our perspective, a more detailed explanation in the introduction, e.g. of the 2D-STE, is not required for the reader. However, we mention this method in our conclusion (s. comment #9). This is the scope of another study we are working on (we will publishing our results soon).

4. The title of Figure 1 should be shortened, and the description can be given within the text.

We followed the guidelines of the journal MDPI diagnostics here: https://www.mdpi.com/journal/diagnostics/instructions. The short caption to figure 1 allows the reader at a glance to understand how the MPI measurement is performed. We advise against an exclusively explanation in the continuous text.

From our perspective the concise caption is needed to understand every single step of the acquisition of Mod-MPI to make the practical handing easier to follow every single step during program application.

5. A flow diagram for the algorithm would be necessary to explain the method.

The software MPI+ is commercially available. So, the algorithm is owned by the manufacturer (Samsung HME). Thus, the details of the intelligent algorithm are also owned by the manufacturer. To understand the algorithm, we included a step-by-step flow chart, which allows the reader to follow the workflow stepwise.

We are in a clinical setting and not in the improving of the underlying artificial intelligence algorithm.

If further information is needed, the white paper can be read: Lee MY, Kim SY, Won HS. White Paper: MPI+, an automatic tool for measurement of fetal right ventricular myocardial performance index. Published online December 9, 2021. https://www.samsunghealthcare.com/en/products/UltrasoundSystem.

6. Results should be tabulated and compared with others from the literature.

The main purpose of this study was to compare the performance of an algorithm between a beginner and an expert So far, the feasibility of Mod-MPI with the software MPI+ has never been investigated between beginners and experts and other comparable algorithms does not exist, and cannot be directly compared. If the reviewer means that the results of the present MPI values should be compared with other studies in the literature, we would like to kindly note that we have already done that in the discussion (line 223).

7. It is recommended to use non-linear fitting to the data in the other figures and compare its accuracy, precision, and sensitivity with the proposed ones https://doi.org/10.3390/bioengineering10020249.

Thank you for this comment. After consultation with our bioinformatic department this is not applicable for our study and our clinical claim. The main purpose of the study was to show the validity of this commercially available algorithm in a clinical setting.

8. The contribution of the work should be highlighted in the discussion and why it is worth publishing.

Thank you for your comment. In the line 184 in the original manuscript we highlighted that our study clearly demonstrates that less experienced operators could benefit from an automated tool for MPI quantification in routine clinical practice. For experts, as well as for beginners, the RV-Mod-MPI is an excellent diagnostic tool for assessment of fetal cardiac function. In total, there was no significant difference between semiautomated measurements of beginner and expert.

For the first time we have demonstrated an advantage of the MPI+ software for a beginner compared to an expert.

In line 268 we mentioned that assessment of MPI could be implemented in routine diagnostics quite simply – effectively on-the-fly (s. #9). The ability of obstetric ultrasonography to detect outflow tract anomalies is increased significantly, if visualization of the ventricular outflow tract complements the four-chamber view well-established in routine diagnostics. As a result of this insight, the outflow tract and the four-chamber view were included in routine screening for CHD, which is evidence-based and consistent with recent guidelines and recommendations from other international expert associations. Its application as a screening as well as follow-up tool for global cardiac dysfunction makes MPI a valuable predictor of CHD.

9. conclusions should be further explored, and a vivid picture of future work is necessary.

We showed that for experts as well as for beginners, the MPI is an excellent diagnostic tool for assessment of fetal cardiac function with the result that assessment of MPI could be implemented in routine diagnostics quite simply. The present values of MPI were very similar to those published by other clinicians who also investigated the clinical value of the MPI+ tool.

We are convinced that implementation of simple methods as those we investigated in our paper may be included in such structured thorough anatomical examinations of the fetal heart and other organs. These will be part of the standard assessment of the fetal state in the future. Such methods give a kind of reassurance for physician and patient. In general, the implementation of intelligent solutions might be of help clinical routine.

We also added the aspect of pathological fetal intrauterine conditions to the wording of the sentence in line 282.

Round 2

Reviewer 2 Report

No Further comments are required.